# LATTE: LATENT ATTENTION FOR LINEAR TIME TRANSFORMERS

## ABSTRACT

The time complexity of the standard attention mechanism in transformers scales quadratically with sequence length. We propose a probabilistic framework for attention, enabling us to derive a novel low-rank linear re-parameterisation of both bidirectional and causal cases, based on defining a latent variable model. Our method can be seamlessly integrated as a drop-in replacement for the standard attention mechanism. Additionally, this framework provides a natural extension for combining local standard attention with our global linear attention. This approach allows us to extend the context length of existing large pre-trained models with only a few additional training steps. The resulting "Latte Transformer" achieves performance comparable to standard attention and other state-of-the-art models, while maintaining linear time and memory complexity, along with constant-time next-token prediction during inference.

## 1 INTRODUCTION

Transformers (Vaswani et al., 2017) are extensively used in sequence modelling, with widespread applications in natural language processing (NLP) (Devlin et al., 2019; Radford et al., 2019; Touvron et al., 2023), and computer vision (Khan et al., 2022; Dosovitskiy, 2020; Liu et al., 2021). The transformer is based on an attention mechanism that compares each element of a sequence with every other. This pairwise interaction gives state-of-the-art results on tasks such as language modelling but it comes at the cost of quadratic time and (in standard implementations) quadratic space complexity. While the quadratic space complexity can be reduced to linear in the sequence length by using non-vectorised operations (Rabe & Staats, 2021), there is no method, with the same properties as the standard attention, to reduce the quadratic time complexity, hindering the application of transformers to very long sequences. Another disadvantage of standard attention is its slow inference for the next token prediction, with time complexity scaling linearly with the length of the conditioning sequence (context window), making it expensive for long sequences.

The attention mechanism takes an input sequence of token vectors $x_1, \ldots, x_T$; $x_s \in \mathbb{R}^D$ and transforms this to a new sequence according to an input-dependent linear function:

$$\tilde{x}_t = \sum_{s=1}^{T} a_{ts} v_s \tag{1}$$

where $a_{ts}$ are the attention weights and $v_t = W_v x_t$ is a linear transformation of the input $x_t$. The standard attention weights are defined using

$$a_{ts} = \frac{\exp\left(q_t^\mathsf{T} k_s\right)}{\sum_{s=1}^{T} \exp\left(q_t^\mathsf{T} k_s\right)} \tag{2}$$

where $k_t = W_k x_t$, $q_t = W_q x_t$ and $W_k, W_q, W_v$ are the key, query and value parameter matrices. In equation 2, we omitted division by the constant factor $\sqrt{d_k}$ given by dimension of $k_t$, for presentation clarity. Since $a_{ts}$ is a positive normalised quantity, we can interpret $a_{ts}$ as the probability $p(s|t)$ of the token occurring at position $s$ given the token occurring at position $t$. It is customary to stack the inputs as row vectors and form the $T \times D$ matrix $X = \text{vcat}\left(x_1^\mathsf{T}, \ldots, x_T^\mathsf{T}\right)$. Similarly, by writing the keys and queries into row-vector stacked matrices, we can write the transformed input sequence in matrix notation as

$$\tilde{X} = \text{Attention}(Q, K, V) \equiv \text{softmax}\left(QK^\mathsf{T}\right)V \tag{3}$$

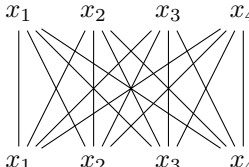 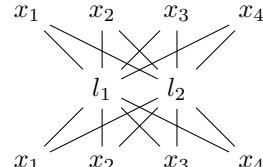

Figure 1: Token-token interaction diagram for non-causal attention of the $p(s|t)$ matrix, see also Lin et al. (2021). The time complexity of each algorithm is $O(ED)$ where $E$ is the number of edges in each graph and $D$ is the dimension of each $x$. (Left) Standard Attention computes all pairwise similarities between the elements of the sequence. (Right) Latte computes only the pairwise similarities between each element of the sequence and each latent state.

In the causal variant, we define $\tilde{x}_t = \sum_{s=1}^{t} a_{ts} v_s$ and the normalisation in equation 2 is replaced with $\sum_{s=1}^{t} \exp\left(k_t^\intercal q_s\right)$ to ensure $\sum_{s=1}^{t} a_{ts} = 1$.

The quadratic cost of attention comes from the matrix multiplication of $Q \in \mathbb{R}^{T \times D}$ and $K^\intercal \in \mathbb{R}^{D \times T}$ in equation 3. The softmax$(\cdot)$ function prevents us from leveraging the associativity of matrix multiplication, resulting in operations involving the matrix $A = \text{softmax}\left(QK^\intercal\right) \in \mathbb{R}^{T \times T}$. To address this and reduce the quadratic complexity, we use a probabilistic interpretation of attention and introduce a latent variable re-parameterisation applicable to both bidirectional and causal cases. Additionally, our framework combines this latent variable model with local sliding window attention, enhancing performance while preserving linear complexity. We also demonstrate how this allows us to extend the context length of a pre-trained large language model with only modest additional compute.

## 2 LATTE ATTENTION

To overcome the quadratic complexity of standard attention we propose Latent Attention (Latte). Instead of comparing the similarity between each token $x_s$ and $x_t$, Latte compares how similar each $x_s$ is with learnt latent tokens, see Figure 1. First, in Section 2.1 we consider a non-causal model before presenting the causal approach in Section 2.2, which is needed for auto-regressive models (Radford et al., 2019). We further show in Section 2.3 how to improve performance by combining a local attention mechanism with global latent attention, which we call Latte Macchiato.

### 2.1 NON-CAUSAL (BIDIRECTIONAL) LATTE

For a sequence of tokens $x_1, \ldots, x_T$, attention, equation 1, transforms a token $x_t$ to

$$\tilde{x}_t = \sum_{s=1}^{T} p(s|t) v_s \tag{4}$$

in which information from the past, present and future tokens is used to define $\tilde{x}_t$. To remove the quadratic dependence of attention on the sequence length, we introduce a latent variable $l$ that conditionally renders attention for token position $s$ independent of token position $t$:

**Definition 1** (Bidirectional Latte). *Let $l$ be a latent variable with $L$ possible states. Under the independence assumption $s \perp\!\!\!\perp t|l$, attention becomes:*

$$\tilde{x}_t = \sum_{s=1}^{T} \sum_{l=1}^{L} p(s,l|t) v_s = \sum_{l=1}^{L} p(l|t) \sum_{s=1}^{T} p(s|l) v_s \tag{5}$$

*Analogous to standard attention we define*

$$p(l|t) = \frac{e^{x_t^\intercal w_l^q}}{\sum_{j=1}^{L} e^{x_t^\intercal w_j^q}}, \quad p(s|l) = \frac{e^{x_s^\intercal w_l^k}}{\sum_{s=1}^{T} e^{x_s^\intercal w_l^k}} \tag{6}$$

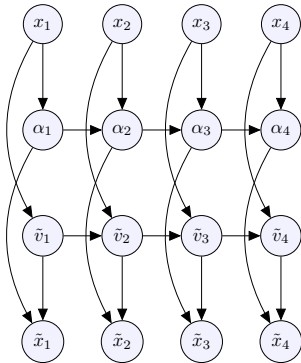

Figure 2: Causal Latte can be written as a recursion in which the variables $\alpha_t = [\alpha_{t,1}, \ldots, \alpha_{t,L}]$ and $\tilde{v}_t = [\tilde{v}_{t,1}, \ldots, \tilde{v}_{t,L}]$ contain all the information required to form the transformed output $\tilde{x}_t$. This recurrent formulation makes a bridge between state-space style recurrent attention approaches and classical attention.

We call $w_l^q$ the $l^{th}$ query vector and $w_l^k$ is the $l^{th}$ key vector – these are learnable parameters of the model. The queries, keys and values are packed into matrices $Q = XW_q$, $K = XW_k$ and $V = XW_v$. Analogous to equation 2, we then define Latte attention as

$$\text{Latte}(Q, K, V) \equiv \underbrace{\text{softmax}_L(Q)}_{T \times L} \underbrace{\text{softmax}_T(K)^{\mathsf{T}}}_{L \times T} \underbrace{V}_{T \times D} \tag{7}$$

where $\text{softmax}_L(\cdot)$ denotes normalisation over the number of latent states and $\text{softmax}_T(\cdot)$ denotes normalisation over the temporal dimension. Latte therefore can also be seen as a rank $L$ parameterisation of the attention matrix. It is important to recognise that we do not aim to approximate the standard attention approach – we rather define a new parameterisation for attention. In our method, it is not meaningful to compute distances between a latent state and a token position, hence we cannot directly use relative positional encodings[1]. To break token position invariance we introduce a new type of relative encoding, VAPOR (Value Acting POsitional Rotations) in Appendix B.4.

## 2.2 CAUSAL LATTE

Causal attention can be written as $\tilde{x}_t = \sum_{s=1}^{t} p(s|t) v_s$, where the distribution $p(s|t)$ is defined by causal attention such that $\sum_{s=1}^{t} p(s|t) = 1$.

**Definition 2** (Causal Latte). *Using a latent variable $l$ with $L$ states, we parameterise causal Latte as:*

$$p(s|t) = \sum_{l=1}^{L} p(s|l, t) p(l|t), \quad p(l|t) = \frac{e^{x_t^{\mathsf{T}} w_l^q}}{\sum_{j=1}^{L} e^{x_t^{\mathsf{T}} w_j^q}}, \quad p(s|l, t) = \frac{e^{x_s^{\mathsf{T}} w_l^k}}{\sum_{s=1}^{t} e^{x_s^{\mathsf{T}} w_l^k}} \tag{8}$$

The dependence on $t$ in $p(s|l, t)$ is due to the causal normalisation up to time $t$. However, this does not imply quadratic time complexity. To see this we define the normalisation terms

$$\beta_t \equiv \sum_{j=1}^{L} e^{x_t^{\mathsf{T}} w_j^q}, \quad \alpha_{t,l} \equiv \sum_{s=1}^{t} e^{x_s^{\mathsf{T}} w_l^k} \tag{9}$$

and write the new representation as

$$\tilde{x}_t = \sum_{l=1}^{L} p(l|t) \sum_{s=1}^{t} p(s|l, t) v_s = \sum_{l=1}^{L} \frac{e^{x_t^{\mathsf{T}} w_l^q}}{\beta_t \alpha_{t,l}} \sum_{s=1}^{t} e^{x_s^{\mathsf{T}} w_l^k} v_s = \sum_{l=1}^{L} \gamma_{t,l} \tilde{v}_{t,l} \tag{10}$$

where

$$\gamma_{t,l} \equiv \frac{e^{x_t^{\mathsf{T}} w_l^q}}{\beta_t \alpha_{t,l}}, \quad \tilde{v}_{t,l} \equiv \sum_{s=1}^{t} e^{x_s^{\mathsf{T}} w_l^k} v_s \tag{11}$$

Causal Latte is therefore also a rank $L$ parameterisation of the attention matrix (see Section 3). Furthermore, since $\tilde{v}_{t,l}$ and $\alpha_{t,l}$ are cumulative sums we can use the recursions

$$\alpha_{t,l} = \alpha_{t-1,l} + e^{x_t^{\mathsf{T}} w_l^k}, \quad \tilde{v}_{t,l} = \tilde{v}_{t-1,l} + e^{x_t^{\mathsf{T}} w_l^k} v_t \tag{12}$$

---

[1]Additive absolute positional encodings can be used as per normal in Latte provided they are applied to the tokens $x_t$, $\tilde{x}_t$, not the latent tokens.

From equation 12 it immediately follows that we can calculate $\tilde{x}_{t+1}$ (*i.e.* infer the future token) directly from $\alpha_{t,l}, \beta_t, \tilde{v}_{t,l}$, whereas standard attention requires the full sequence $x_1, \ldots, x_t$. In this sense, Causal Latte is a recurrent model, similar in essence to Recurrent Neural Networks (RNNs) and state space models (SSMs) (Gu et al., 2021; Smith et al., 2022; Fu et al., 2023), see Figure 2. A numerically stable implementation of the recursion is given in Appendix B.3. Next token inference in Causal Latte requires $O(LD)$ memory, compared to $O(TD)$ in standard attention. Similarly, the time complexity is $O(LD)$, compared to $O(TD)$ in standard attention. Next token inference is therefore significantly faster than standard attention, assuming $L \ll T$. For training we can calculate all terms $\tilde{x}_1, \ldots, \tilde{x}_T$ in total time $O(TLD)$ and total space $O(TL + LD)$. This is compared to $O(T^2 D)$ time and $O(TD)$ space complexity for standard attention (Rabe & Staats, 2021). Table 9 in Section B.2 summarises the complexity compared to other efficient attention approaches, see Section 3.

## 2.3 LATTE MACCHIATO

Whilst Latte uses latent states to represent global concepts and share long-range information across a sequence, it may not account for local information as effectively as standard attention since it lacks elementwise comparisons. Therefore combining linear attention with standard attention is natural and has been explored in works such as Hua et al. (2022); De et al. (2024). Different to prior models in our work we combine local and global context by a simple extension of our latent variable model.

**Definition 3** (Latte Macchiato). *Let $l = 0$ be a special latent state allocated to standard attention. We define Latte Macchiato as a weighted mixture of standard attention and Causal Latte:*

$$\tilde{x}_t = p(l = 0|t) \sum_{s=1}^{t} p_0(s|t) v_s + \sum_{l=1}^{L} p(l|t) \sum_{s=1}^{t} p(s|l, t) v_s \tag{13}$$

Here $p_0(s|t) \equiv p(s|l = 0, t)$ represents standard attention; in practice to retain computational tractability we use sliding window attention with a window size $w$:

$$p_0(s|t) = \begin{cases} \dfrac{e^{q_t^\mathsf{T} k_s}}{\sum_{s=t-w}^{t} e^{q_t^\mathsf{T} k_s}} & t - w \leq s \leq t \\ 0 & otherwise \end{cases} \tag{14}$$

where we now define $p(l|t)$ to ensure normalisation over all $L + 1$ states, $l = 0, \ldots, L$.

## 2.4 EXTENSIONS

In Definition 3 of Latte Macchiato, the quantity $p(l|t)$ depends solely on the token $x_t$, while $p(s|l, t)$ is based only on $x_s$ from the entire sequence. To encourage the latent states to capture temporal dependencies across multiple sub-words, we use a 1D convolution of size $K$ to compute these probabilities:

$$y_t = \sum_{i=0}^{K} w_i^c x_{t-i}, \qquad p(l|t) = \frac{e^{y_t^\mathsf{T} w_l^q}}{\sum_{j=1}^{L} e^{y_t^\mathsf{T} w_j^q}}, \qquad p(s|l, t) = \frac{e^{y_s^\mathsf{T} w_l^k}}{\sum_{s=1}^{t} e^{y_s^\mathsf{T} w_l^k}} \tag{15}$$

We observed that performance improves with larger convolution sizes $K$ prompting us to also extend $y_t$ to depend on all previous tokens using a linear recurrent neural network. For our experiments, we used the recurrent gated linear unit (RGLRU) layer introduced by De et al. (2024) which, compared to a convolution, is also input dependent. See Figure 3 for an overview of the architecture. Note that both the convolution and recurrent layers break positional invariance, thereby eliminating the need for VAPOR positional encodings in these extensions.

## 3 RELATED WORK

The literature surrounding efficient attention is extensive, but can be broadly classified into six overlapping classes (Tay et al., 2022): Downsampling Jaegle et al. (2021), Random Patterns (Zaheer et al., 2020), Learnable Patterns Wang et al. (2022); Kitaev et al. (2019), Sparse Transformers (Ainslie et al., 2020; Beltagy et al., 2020), Recurrence (Dai et al., 2019) and Low-Rank (Wang et al., 2020;

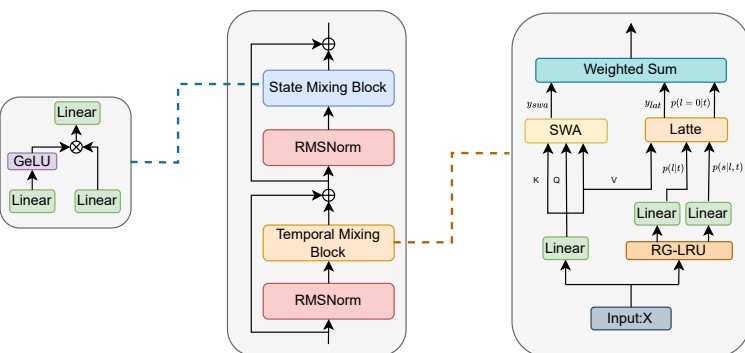

Figure 3: One layer in the architecture of Latte Macchiato (Latte-RGLRU-SWA++).

Katharopoulos et al., 2020). Other papers focus on linear-attention and use special hardware optimisations Qin et al. (2024b); Sun et al. (2024). Here we provide an overview of the most related work on addressing attention in transformers. See also Lin et al. (2021).

**Efficient Transformer.** Shen et al. (2021) proposed an equivalent bidirectional model to equation 7 but did not explore the latent variable interpretation that allows for a natural extension to the important case of casual attention. Our method also easily integrates with standard local attention models. Furthermore, they experiment with vision tasks, while our work focuses on language problems.

**Luna.** Linear Unified Nested Attention (Ma et al., 2021) performs attention between $T$ input tokens and a sequence of $L$ latent tokens. For the bidirectional case, Luna uses nested attention

$$Y = \text{softmax}(\underbrace{Q}_{L \times D} \underbrace{K^\mathsf{T}}_{D \times T}) \underbrace{V}_{T \times D}, \qquad \tilde{X} = \text{softmax}(\underbrace{Q'}_{T \times D} \underbrace{K'^\mathsf{T}}_{D \times L}) \underbrace{V'}_{L \times D} \tag{16}$$

where $Q = W_q P$, $K = W_k X$, $V = W_v X$, $Q' = W_q' X$, $K' = W_k' Y$, $V' = W_v' Y$ and $P$ is a model parameter representing the sequence of $L$ latent tokens. This differs substantially from our formulation in equation 7. For causal attention one needs to modify the approach such that for each token $x_t$ the latent states only interact with $x_{\leq t}$. The Luna causal layer can be written as:

$$\tilde{x}_t = \frac{1}{t} \sum_{l=1}^{L} B_{t,l} \sum_{s=1}^{t} A_{l,s} x_s \tag{17}$$

Whilst this matrix factorisation looks similar to causal Latte equation 10, $B_{t,l}$ and $A_{l,s}$ are parameterised differently. For $A$, Luna uses *softplus* (Glorot et al., 2011) and *elu* (Clevert et al., 2015) non-linearities and $B$ depends on $A$:

$$A = \zeta \text{elu}\left(P X^\mathsf{T}\right) + 1 \in \mathbb{R}^{L \times T} \qquad B_{t,:} \propto \exp\left(x_t^\mathsf{T} \frac{1}{t} \sum_{s=1}^{t} x_s A_{:,s}^\mathsf{T}\right) \in \mathbb{R}^{T \times L} \tag{18}$$

In equation 18, each $B_t$ is normalized by summing over the $L$ latent states. While Luna shares a similar motivation to Latte in terms of leveraging latent states, its implementation differs substantially. Additionally, their model does not incorporate local standard attention.

**Recent Work.** Recent studies have investigated state-based models for mixing tokens within a sequence. Although inspired by first-order differential equations, these models can also be viewed as linear recurrent neural networks (Gu et al., 2021; Smith et al., 2022). Gated sequence models like Mamba (Gu & Dao, 2023) improve the performance of these models on text by introducing input-dependent weights in the recurrence. In contrast, other approaches combine local attention mechanisms with recurrent layers (Ma et al., 2023; De et al., 2024; Zuo et al., 2022). Our method aligns more closely with the latter class of models, but differs in how we integrate local attention with global linear context. Furthermore, using our Latte-Macchiato method we can easily extend the context size of a large pre-trained model in linear time.

Table 1: Iterative improvement of Latte language modelling. SWA: Sliding Window Attention.

| Model | Params. | PPL ↓ |
|---|---|---|
| Latte | 111M | 21.88 |
| Latte++ | 140M | 21.56 |
| Latte-Conv++ | 140M | 20.26 |
| Latte-Conv-SWA++ | 153M | 18.52 |
| Latte-RGLRU++ | 139M | 19.99 |
| Latte-RGLRU-SWA++ | 153M | **17.64** |
| Transformer++ | 151M | **17.19** |

Table 2: Comparison of Latte-Macch with other linear-scaling models on language modelling. SWA: Sliding Window Attention

| Model | Params. | PPL ↓ |
|---|---|---|
| Mega Ma et al. (2023) | 153M | 23.75 |
| Retnet Sun et al. (2023) | 197M | 21.59 |
| H3 Fu et al. (2023) | 125M | 21.0 |
| RWKV Peng et al. (2023a) | 153M | 18.97 |
| Griffin De et al. (2024) | 139M | 18.83 |
| RGLRU-SWA++ (our) | 141M | 18.25 |
| Mamba Gu & Dao (2023) | 149M | 17.70 |
| GLA Yang et al. (2024) | 206M | 19.10 |
| Ligth.Att Qin et al. (2024a) | 166M | 23.67 |
| Latte-RGLRU-SWA++ | 153M | **17.64** |

## 4 EXPERIMENTS

We implement Latte using Jax (Bradbury et al., 2018), which enables efficient handling of the recurrence operation equation 12 via the `scan` operator[2]. In Section 4.1, we analyse how each Latte component improves the model, followed by a comparison with state-of-the-art (SOTA) models. Section 4.2 covers Latte's runtime, and Section 4.3 provides an analysis of its latent states and sequence extrapolation properties. Section 4.4 presents results on LRA data, while Section 4.5 discusses the benefits of using large pre-trained models with Latte Macchiato.

### 4.1 LANGUAGE MODELLING

We use OpenWebText (Gokaslan & Cohen, 2019) data to train small models on the next-token prediction task. To ensure a fair comparison, we train all models in this section under the same settings for 8 billion tokens. For a complete overview of data pre-processing and hyperparameters, see Appendix A.1 and Table 6.

**Base model.** Table 1 compares perplexity (PPL) across architectures. The first model (Latte) uses Latte layers for temporal mixing, as in Definition 2, with VAPOR positional encodings (Appendix B.4). The rest of the architecture follows the standard transformer (Vaswani et al., 2017). In the second model, we replace layer normalisation with RMSNorm (Zhang & Sennrich, 2019) and the feedforward network with a Gated Linear Unit (GLU) (Dauphin et al., 2017), referring to this version as "++" (e.g., Transformer++, Latte++). The change results in a marginal performance improvement.

**Convolution and sliding window attention.** We enhance Latte++ with a short convolution layer ($K = 3$), as detailed in Section 2.4, yielding significant performance gains with minimal parameter increase. Next, we add a sliding window attention (SWA) of 128 tokens to create Latte-Macchiato, which increases parameters but delivers a substantial performance boost. For the SWA we use ROPE positional encodings (Su et al., 2024).

**Recurrent layer and sliding window attention.** Replacing the 1D convolution with a Recurrent Gated Linear Unit (RGLRU) further improves performance and slightly reduces parameters. Finally, adding SWA with ROPE positional encoding to Latte-RGLRU++ results in our best model, combining a 128-token SWA window and global Latte attention for significant gains.

**Comparison to state-of-the-art models.** We further compare our model with other linear-time ("efficient") models in Table 2. To show that not only SWA and RGLRU, but also the Latte components contribute to reducing perplexity, we introduce a competing model (RGLRU-SWA++). This model removes the Latte components, constructing the queries and keys for sliding window attention using the normalized outputs of the RGLRU layer: $Y = \text{RGLRU}(X)$, $Z = \text{RMSNorm}(Y)$, $Q = W_q Z$, $K = Z W_k$, and $V = X W_v$. This approach is conceptually similar to that of the Mega

---

[2]PyTorch would require a custom CUDA kernel, as `for` loops are inefficient even when compiled.

model (Ma et al., 2023). Our model achieves the best performance, despite a slight increase in the number of parameters. Although it is difficult to match parameter counts across all models, we kept the number of layers, hidden units, and feed-forward dimensions consistent. Additionally, Griffin's window attention size was set to 128 to align with Latte-Macchiato. All other hyperparameters are detailed in Table 6.

These results show that combining standard sliding window attention with global latent tokens (Latte-Macchiato) is competitive with SOTA in language modelling. Another important fact of Latte is that it can build global sequence information directly on top of a pre-trained language model, allowing a pre-trained model to be applied to much longer contexts, see also Section 4.5.

## 4.2 EMPIRICAL RUNTIME EFFICIENCY

We compare the runtime of a forward pass for both the convolutional (Latte-Conv-SWA++) and recurrent (Latte-RGLRU-SWA++) variants of Latte Macchiato against standard attention mechanisms across different sequence lengths and model sizes. To ensure a fair comparison, we use the same hyper-parameters and adjust the batch size with sequence length to maintain a constant number of tokens. As illustrated in Figure 4, both versions of Latte Macchiato outperform standard attention in terms of speed. Given the performance improvements of Latte-RGLRU-SWA++ over Latte-Conv-SWA++ for only a modest time increase, the recurrent approach seems generally preferable to the convolution approach, presumably since longer-range information is taken into account. While our work has concentrated on reducing the theoretical time and memory complexity of attention, future optimisation at the kernel level could further enhance practical runtime performance.

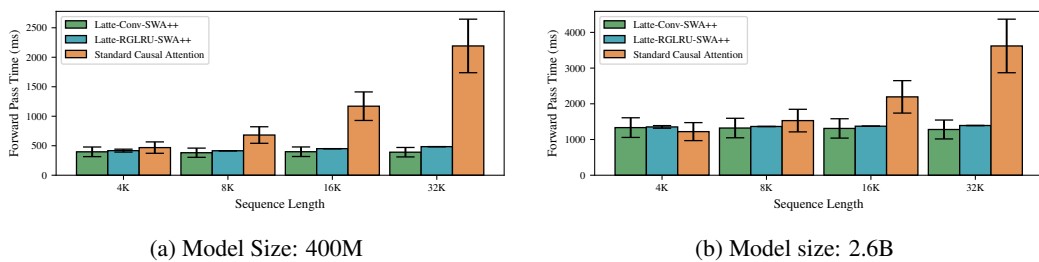

(a) Model Size: 400M          (b) Model size: 2.6B

Figure 4: Runtime in milliseconds (ms) for forward passes at different sequence lengths and model sizes. Measurements are repeated for 100 runs, and the plot displays the standard deviation. While the standard deviations for Latte-RGLRU-SWA++ are included, they are too small to be visible.

## 4.3 LATENT STATE ANALYSIS

**Impact of the state size.** Firstly, we look at the performance of Latte-RGLRU++ as we increase the number of latent states $L$. Note that we do not use Latte-RGLRU-SWA++ (Latte Macchiato) in order to eliminate any confounding factor introduced by local attention; we set the number of heads to 1 for the same reason. Models are trained on 0.6 billion tokens with sequences of length 1024 from OpenWebText data. The same pre-processing is followed as in Section 4.1. In Figure 5a we see that increasing $L$ improves the performance; nevertheless there is a point at which the added computational cost of using more latent states is no longer justified by the marginal performance gains. As expected, standard causal attention on the full context performs better.

**Latte-Macchiato vs. Sequence Truncation.** For long sequences where standard attention's quadratic complexity is too expensive, we compare the common sequence truncation approach to our linear Latte-Macchiato layer (Latte-RGLRU-SWA++). Latte-Macchiato uses a 128-token sliding window on the full sequence. Figure 5b shows Latte-Macchiato outperforming standard attention on truncated sequences. We maintain a consistent number of training tokens across sequence lengths by adjusting batch sizes for standard causal attention. Both methods use 128 latent states and identical hyperparameters, see Table 6 for more details. We use BookCorpus dataset (Zhu et al., 2015) because it has longer raw sequences than OpenWebText, thus making global information more relevant.

**Sequence Extrapolation.** Considering that long sequences improve perplexity, we also examine the model's ability to extrapolate to sequences longer than those seen during training. Specifically,

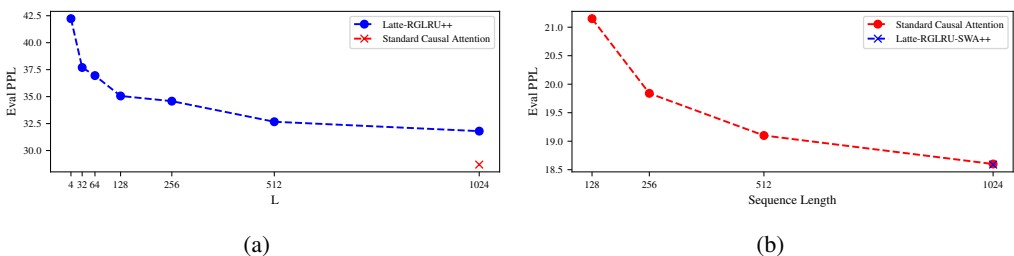

(a)                                                                    (b)

Figure 5: (a) The effect of latent state size on test perplexity for 1024-token sequences in Latte-RGLRU++. Compared with standard causal attention. (b) The impact of increasing sequence length on the performance of standard causal attention. Compared with RGLRU-SWA++ using 128 long sliding window and trained on 1024 sequence length.

we train on 5K-token sequences from the BookCorpus dataset and evaluate on sequences up to 16K tokens. A comprehensive overview of the training hyperparameters is provided in Table 6. As shown in Figure 6, our model successfully extrapolates to longer sequences, whereas the performance of standard causal attention degrades as the sequence length increases. Notably, Latte-RGLRU-SWA++ also achieves performance comparable to standard attention on sequences seen during training. We use YARN (Peng et al., 2023b) relative positional encoding for standard attention and ROPE for sliding window attention in Latte. YARN is used because it helps transformers extrapolate.

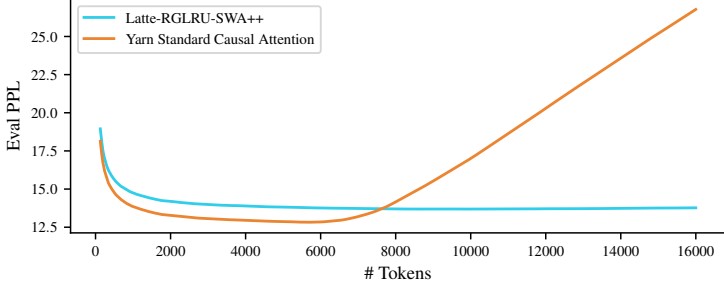

Figure 6: Sequence Length Extrapolation for Attention and Latte-Macch. Both models were trained on sequences of length 5K which are extrapolated to 16K during testing.

**Latent Collapse.** A common issue with latent variable models is latent collapse, where only a small subset of latent states is used. As demonstrated in Figure 7, Latte does not exhibit latent collapse, even in the absence of dropout. The figure also highlights that local attention and Latte attention are effectively used, with the probability mass distributed across various latent states. The plots are generated from various heads and layers of the Latte-RGLRU-SWA++ model, as detailed in Table 2. We provide plots for all layers and heads in Figure 14.

## 4.4 BIDIRECTIONAL TASKS

Long-Range Arena (LRA) (Tay et al., 2021) is a collection of classification benchmark designed to evaluate the long-range capabilities of models, with context lengths ranging from 2K to 16K tokens (see Appendix A.2). All models used in these experiments are bidirectional. Since Latte shares more similarities with transformer architectures, we primarily compare it against transformer baselines. However, it is important to note that state-based models currently represent the state-of-the-art on LRA tasks (Gu et al., 2021; Smith et al., 2022).

Our initial model, Latte, implements the bidirectional formulation introduced in Definition 1 and employs absolute positional encodings. As shown in Table 3, this model performs comparably to the best transformer-based competitors, but it does not reach the performance levels of time-invariant state space models. However, when we incorporate the recurrent linear layer, performance improves significantly for the discrete tasks in the benchmark, although the image datasets continue

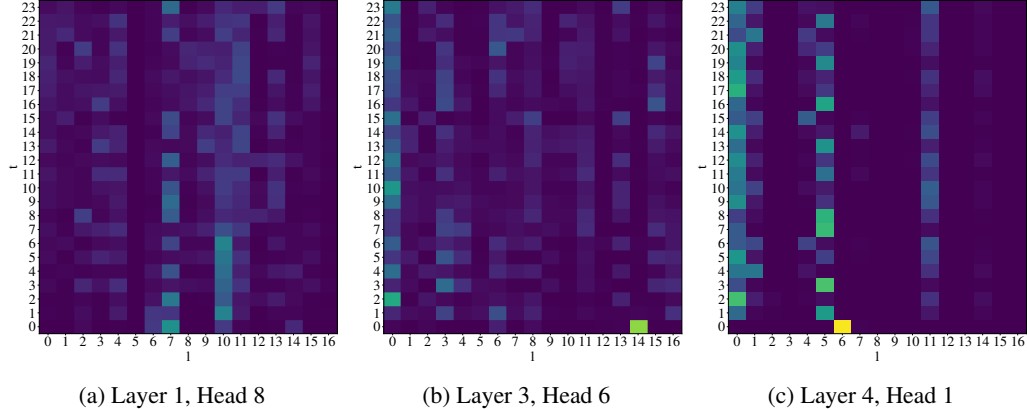

| (a) Layer 1, Head 8 | (b) Layer 3, Head 6 | (c) Layer 4, Head 1 |

Figure 7: Latte Macchiato. Plots of $p(l|t)$ for different layers and heads across a sequence of 25 tokens and $l = 0, \ldots, 16$. State 0 corresponds to using standard causal windowed attention, whereas states higher than zero correspond to global latent tokens. Brighter means higher probability. Hyperparameters are provided in Table 6.

Table 3: Classification accuracies for LRA dataset. We report the best test score (higher is better). All Latte versions are bidirectional.

| Model | ListOps | Text | Retrieval | Image | Pathfinder |
|---|---|---|---|---|---|
| Bid. Attention | 36.37 | 64.27 | 57.46 | 42.44 | 71.40 |
| Linformer | 35.70 | 53.94 | 52.27 | 38.56 | 76.34 |
| Performer | 18.01 | 65.40 | 53.82 | 42.77 | 77.05 |
| Longformer | 35.63 | 62.85 | 56.89 | 42.22 | 69.71 |
| Reformer | 37.27 | 56.10 | 53.40 | 38.07 | 68.50 |
| BigBird | 36.05 | 64.02 | 59.29 | 40.83 | 74.87 |
| Linear Transformer | 16.13 | 65.90 | 53.09 | 42.34 | 75.30 |
| Luna Bid. $L = 128$ | 38.01 | 65.74 | 79.55 | 47.47 | 78.89 |
| Mega-Chunk | 58.76 | 90.19 | 90.97 | 85.80 | 94.41 |
| S5 | 62.15 | 89.31 | 91.40 | 88.00 | 95.33 |
| Latte | 40.18 | 64.51 | 73.39 | 47.55 | 75.61 |
| Latte-RGLRU++ | 56.7 | 83.85 | 81.07 | 57.61 | 72.13 |
| Latte-RGLRU-SWA++ | 61.39 | 85.8 | 87.67 | 70.19 | 73.69 |

to lag behind. This is a common trend, where even state space models like Mamba, which excel in discrete tasks such as language modelling, perform poorly on continuous data[3]. Performance on discrete tasks is further enhanced by using bidirectional sliding window in Latte-Macchiato (Latte-RGLRU-SWA++). The full set of hyperparameters is provided in Table 8.

## 4.5 EXTENDING A PRE-TRAINED MODEL

Training large models from scratch is computationally expensive, even when the sequence mixing layer (*e.g.*, attention) has linear time and memory complexity. Recent work has demonstrated preliminary success in distilling pre-trained quadratic self-attention layers into sub-quadratic layers, such as Mamba (Bick et al., 2024). However, unlike Latte, these architectures (Gu & Dao, 2023) significantly differ from the attention mechanisms used in standard transformers, making knowledge distillation from pre-trained transformers more complex. Other research has modified relative embeddings in standard attention to enable sequence extrapolation, but the computational cost remains quadratic (Sun et al., 2022). We use Latte-Macchiato with SWA weights taken from a pre-trained large model and show that training only the Latte-specific weights for 1.6B tokens is sufficient.

---

[3]As discussed by the authors of Mamba in github.com/state-spaces/mamba/issues/282

This approach enables us to achieve desirable properties, such as global context and effective sequence length extrapolation, by bootstrapping from a pre-trained open-source large language model.

In our experiments, we use a pre-trained 2.6B Gemma model (Gemma-Team, 2024) and replace the standard attention layer with a Latte-Macchiato layer of 128 long sliding window attention. The model is trained on the SlimPajama dataset (Soboleva et al., 2023), for a single day on four 80GB A100 GPUs, see Table 7 for more details. In Table 4, we evaluate both the original Gemma model and our modified version, Gemma Macchiato, on the validation set as well as other publicly available corpora[4]. First, on sequences of length 4K, which match the training length, we find that our model's results are comparable to or even exceed those of the original model. When extending the sequence length to 8K and 16K tokens, our model significantly outperforms Gemma, demonstrating that excellent context extrapolation capabilities are acquired with minimal additional training steps.

Table 4: PPL ↓ on the validation set for 4K, 8K and 16K sequences. Gemma-Macchiato is initialised from Gemma and pre-trained on Slim-Pajama. Unlike Gemma-Macchiato, Gemma fails to generalise to longer sequences.

| Dataset | Gemma | | | Gemma-Macchiato | | |
|---|---|---|---|---|---|---|
| | 4K | 8K | 16K | 4K | 8K | 16K |
| Slim-Pajama | 10.97 | 36.35 | 294.18 | 10.14 | 9.99 | 10.27 |
| Pile | 7.42 | 19.26 | 243.54 | 7.27 | 6.98 | 7.04 |
| OWT | 10.75 | 38.36 | 252.74 | 10.76 | 10.72 | 10.99 |
| Tiny-Stories | 5.45 | 19.15 | 66.61 | 4.26 | 4.34 | 4.30 |

We also check the abilities of the distilled model on a standard natural language harness of multiple-choice question-answering. Like the general trend of linear models, performance decreases especially on tasks like MMLU Mercat et al. (2024); Zhang et al. (2024). However, our aim is not to outperform standard attention, but to provide a linear global context extension method which is a better alternative to sequence truncation, often when the quadratic cost of attention becomes a limitation.

Table 5: Common Few Shot learning benchmarks. The score is accuracy or normalized accuracy (↑). We use a sliding window of size 128.

| Model | MMLU | HellaSwag | Lambada | ARC-C | ARC-E | WinoG | Piqa | BoolQA |
|---|---|---|---|---|---|---|---|---|
| Gemma2 2B | 53.0 | 73.03 | 69.8 | 53.4 | 80.2 | 71.4 | 79.1 | 73.61 |
| Gemma-Mach 2B | 46.8 | 73.11 | 68.29 | 52.9 | 76.9 | 70.6 | 78.7 | 71.48 |
| Mamba (3B) | 26.2 | 71.0 | - | 41.7 | 68.2 | 65.9 | 78.1 | 71.0 |
| GLA (1.3B) | - | 49.8 | 46.9 | 26.6 | 57.2 | 53.9 | 71.8 | - |

## 5 CONCLUSION

We introduced a latent attention mechanism that scales linearly with sequence length and serves as a drop-in replacement for standard attention. While previous approaches have explored low-rank representations, to the best of our knowledge, none have interpreted these methods as a latent variable model, from which a straightforward formulation for both bidirectional and causal variants can be derived, giving SOTA results in language modelling. Furthermore, our framework extends this approach by integrating local sliding window attention with global latent attention. This allows one to now easily take a pre-trained large language model and considerably extend its usable context length with only modest additional training and at minimal additional run-time cost.

Our initial focus has been on language modelling and long-range classification tasks and future work will consider tasks such as question-answering and extensions to multimodal models.

---

[4]We use the version without copyrighted content huggingface.co/datasets/monology/pile-uncopyrighted

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

## A  EXPERIMENTAL DETAILS

This section describes in detail the datasets and hyperparameters used for our language modelling and classification experiments.

### A.1  LANGUAGE MODELLING

OpenWebText (Gokaslan & Cohen, 2019) is an open-source version of the corpus used to train GPT (Radford et al., 2019) and consists of 41 GB of data extracted from 8,013,769 documents. We tokenize the corpus using a pre-trained Byte Pair Encoding (BPE) tokenizer with a vocabulary of 50,267 tokens. We also ensure that sequences are consistently of length 1024 by concatenating consecutive tokenized examples until this length is reached. This eliminates the need for padding, ensuring that it is not a confounding factor and results in a more efficient computation.

#### A.1.1  HYPERPARAMETERS

In this section, we describe the hyper-parameters used in all of the language modelling experiments. Where the hyper-parameter is missing, it means that we vary it in the experiment and its value is clear from the corresponding section in the paper.

### A.2  LRA DATASET

This section displays the hyperparameters employed in the bidirectional experiments and provides a brief description of the synthetic datasets utilized in the LRA corpus. A more comprehensive account can be found in the original paper (Tay et al., 2021).

In the experiments, one layer consists of a standard transformer block where the transformation operation that gives $\tilde{x}_t$ is Latte or Standard Attention. For positional encoding, we use the classic transformer sinusoidal embeddings. This convention holds for both bidirectional and unidirectional problems. A complete implementation can be found in our code repository: "dummy_url".

#### A.2.1  LISTOPS

The dataset contains sequences up to length 2048 of numbers from 0 to 9 and four operators: *MAX, MEAN, MEDIAN* and *SUM_MOD*. Parentheses are used to delimit the reach of each operator. The answer is also a digit from 0 to 9, which allows us to easily transform the problem into a ten-way classification task.

Table 6: Hyperparameters for the language generation task. $LR$ is the learning rate and "#" denotes "the number of".

| HyperParam. | OpenWebText | Hyperparams. Figure 5 | Hyperparams. Figure 6 |
|---|---|---|---|
| $\#Layers$ | 12 | 12 | 12 |
| $\#Heads$ | 8 | 12 | 8 |
| Hidden Dim ($D$) | 756 | 512 | 512 |
| Feed Forward Dim. | 3072 | 2048 | 2048 |
| Latent Dim ($L$) | 128 | 128 | 128 |
| Local Attention Window | 128 | 128 | 128 |
| Convolution Kernel ($K$) | 3 | 3 | 3 |
| Dropout | 0.0 | 0.0 | 0.2 |
| $LR$ | 0.0006 | 0.0006 | 0.00025 |
| $LR$-Warmup | 2000 | 1000 | 2000 |
| $LR$-Decay | Cosine | Cosine | Cosine |
| $\#Iters.$ | 100000 | 10000 | 300000 |
| Weight Decay | 0.01 | 0.01 | 0.01 |
| Seq. Len. ($T$) | 1024 | - | 5000 |
| Batch Size ($B$) | 80 | - | 4 |
| Tokenizer | BPE(57K) | BPE(57K) | BPE(57K) |
| Embedding Type | Rope | Rope | Rope |
| Unroll Factor | 256 | 256 | 256 |

Table 7: Hyperparameters for adopting Gemma to our framework. All the Gemma hyperparameters are kept intact: huggingface.co/google/gemma-2-2b. $LR$ is the learning rate and "#" denotes "the number of".

| HyperParam. | Value |
|---|---|
| Local Attention Window | 128 |
| Latent Dim ($L$) | 128 |
| $LR$ | 0.0006 |
| $LR$-Warmup | 2000 |
| $LR$-Decay | Cosine |
| $\#Iters.$ | 100000 |
| Seq. Len. ($T$) | 4096 |
| Batch Size ($B$) | 4 |
| Tokenizer | Gemma2 |

Table 8: Hyperprameters used for training on LRA. Number of latent states $L$ specified in the result table. $H$=number heads, $D$=hidden dimension, $LR$=learning rate, $B$=batch size, $WD$=weight decay. $\#Layers$ denotes the number of layers which include attention/approximation of attention and non-linear projections. "Embed." is the type of embedding used by the SWA.

| Dataset | $\#Layers$ | $H$ | L | $D$ | $LR$ | $B$ | $WD$ | Dropout | Epochs | Embed. |
|---|---|---|---|---|---|---|---|---|---|---|
| ListOps | 6 | 4 | 40 | 128 | 1e-3 | 64 | 0.01 | 0.1 | 50 | Rope |
| Text | 6 | 4 | 256 | 256 | 1e-3 | 32 | 0.05 | 0.1 | 32 | Rope |
| Retrieval | 6 | 4 | 40 | 128 | 1e-4 | 32 | 0.01 | 0.1 | 20 | Rope |
| Image | 6 | 4 | 40 | 512 | 1e-3 | 32 | 0.05 | 0.1 | 200 | Absolute |
| Pathfinder | 6 | 4 | 256 | 256 | 1e-3 | 64 | 0.03 | 0.2 | 200 | Absolute |

### A.2.2 TEXT

The Text corpus is represented by a binary classification task with long text sequences. One can easily obtain large contexts from existent datasets by tokenizing at the character level. This part of the benchmark is derived from the IMDb (Maas et al., 2011) movie review corpus, resulting in 4K character sequences.

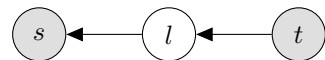

Figure 8: Graphical model for bidirectional Latte, $l$, $t$ and $s$ are discrete random variables.

### A.2.3 RETRIEVAL

This dataset tests the ability of a model to predict the similarity between two long documents. Similarly to the previous corpus, it ensures long contexts through character-level tokenization, resulting in 4K tokens per document. Using a "two-tower model setup" (Tay et al., 2021) the total sequence length becomes 8K. This is a binary classification problem, which uses accuracy as a metric.

### A.2.4 IMAGE

Alongside text, images can also exhibit long-range dependencies by flattening the original image into a sequence. The Image dataset is the sequential version of Cifar10 (Krizhevsky et al., 2009), which contains images of 10 different entities: "airplane, automobile, bird, cat, deer, dog, frog, horse, ship, truck". To obtain a sequence with one input channel we apply a grayscale transformation. The model needs to predict the correct entity class, given the flattened image represented as a sequence of tokens.

### A.2.5 PATHFINDER

This part of the benchmark is also represented by images where the task is to predict whether there is a path between two points in a black-and-white image. This dataset consists of $32 \times 32$ images which after flattening result in sequences of length 1024. In general larger sequences can be created by increasing the resolution. Data is tokenized similarly to the image dataset in Section A.2.4.

### A.2.6 PATHX

This dataset is a version of PathFinder where the image size is increased to $128 \times 128$, resulting in flattened sequences of 16384 tokens. Since all the transformer architectures fail on this dataset, we do not add it to the benchmark.

## B LATTE

### B.1 GRAPHICAL MODEL

Our latent variable model consists of three random variables as depicted in Figure 8.

### B.2 TIME AND MEMORY COMPLEXITY

For the training complexity on a sequence of length $T$, we note that given $\alpha_{t-1,l}$, we need $O(LD)$ time and $O(L)$ memory to compute $\alpha_{t,l}$. Since each $\beta_t$ requires only $O(LD)$ time and $O(1)$ space, all the terms $\gamma_{t,l}$ can be precomputed using matrix multiplication in time $O(TLD)$ and $O(TL)$ storage[5]. Similarly, given $\tilde{v}_{t-1,l}$ one needs $O(LD)$ time to calculate the update to $\tilde{v}_{t,l}$ and $O(LD)$ space to (in-place) store the update $\tilde{v}_{t,l}$. Equation 10 therefore allows us to calculate all terms $\tilde{x}_1, \ldots, \tilde{x}_T$ in total time $O(TLD)$ and total space $O(TL + LD)$. This is compared to $O(T^2D)$ time and $O(TD)$ space complexity for standard attention[6] (Rabe & Staats, 2021).

---

[5]$\gamma_{t,l}$ can be computed on the fly for each $\tilde{x}_t$ reducing the memory complexity to $O(L)$. However, we precompute it to take advantage of matrix multiplication on GPU.

[6]This is based on a sequential implementation – using a naive vectorised GPU implementation retains the time complexity but increases the space requirement to $O(TLD)$ for Latte and $O(T^2D)$ for standard attention.

Table 9: Time and memory complexity during training. Only dominating terms are kept. $T$ - sequence length, $L$ - number of latent states, $D$ - hidden dimension $W$ - local attention window.

| Model | Time | Memory |
|-------|------|--------|
| Latte Causal | $O(TLD)$ | $O(TLD)$ |
| Latte Causal Seq. | $O(TLD)$ | $O(TL + LD)$ |
| Latte Machiatto | $O(TLD + TWD)$ | $O(TLD + TWD)$ |
| Latte-Machiatto Seq. | $O(TLD + TWD)$ | $O(TL + LD + TWD)$ |
| Latte Bidirectional | $O(TLD)$ | $O(TL + LD)$ |
| Attention | $O(T^2D)$ | $O(T^2D)$ |
| Seq. Attention | $O(T^2D)$ | $O(TD)$ |
| Linformer | $O(TLD)$ | $O(TLD)$ |
| Perceiver | $O(TLD)$ | $O(TLD)$ |

### B.3 NUMERICAL STABILITY

The term $p(l|t)$ in equation 8 can be calculated in a numerically stable way using exponential-max-normalisation[7]. However, we cannot directly apply the same approach to stabilise $\alpha_{t,l}$ since we require $p(s|l,t)$ in equation 8 to normalise for each $t$; furthermore, this must be computed sequentially to retain the optimal computational complexity. To exemplify this difference, consider a sequence $y = [1, 10, 1000]$, for which we require normalised distributions $e^{y_i} / \sum_{j=1}^{i} e^{y_j}$ for each element $i \in \{1, 2, 3\}$. In this case, subtracting the maximum value 1000 from each element $[-999, -990, 0]$ will result in underflow for the first two elements when exponentiated, giving numerically meaningless results, except for the third distribution. We address this using a running maximum approach (Rabe & Staats, 2021). Let $\theta_{t,l} = x_t^\intercal w_l^k$ and $\theta_{t,l}^* = \max_{s \in \{0,...,t\}} \theta_{s,l}$. We then use a sequential computation:

$$\alpha_{t,l} = \alpha_{t-1,l} e^{\theta_{t-1,l}^* - \theta_{t,l}^*} + e^{\theta_{t,l} - \theta_{t,l}^*}, \qquad \tilde{v}_{t,l} = \tilde{v}_{t-1,l} e^{\theta_{t-1,l}^* - \theta_{t,l}^*} + e^{\theta_{t,l} - \theta_{t,l}^*} v_t \qquad (19)$$

This value for $\alpha$ is then used to define $\gamma$ in equation 10, which is in turn used with $\tilde{v}$ above to compute $\tilde{x}_t$.

### B.4 RELATIVE EMBEDDINGS

Relative embeddings have the advantage of generalising to sequence lengths unseen during training. Since Latte calculates similarities between sequence tokens and global latent states, for which it is less meaningful to define a relative distance. We therefore introduce VAPOR (Value Acting POsitional Rotations) that computes values based on the relative distance between tokens, but leaves the attention weights unaffected based on introducing a learnable rotation matrix $R$:

$$\tilde{x}_t = \sum_{s=1}^{t} p(s|t) R^{t-s} v_s = R^t \sum_{s=1}^{t} p(s|t) R^{-s} v_s = R^t \sum_{l=1}^{L} p(l|t) \sum_{s=1}^{t} p(s|l,t) R^{-s} v_s \qquad (20)$$

Notice that in equation 20 we split the power of the rotation matrix in $R^t$ prefactor and $R^{-s}$ postfactor. This is computationally advantageous since for Latte we can reuse the calculation of the inner sum for different tokens $t$. Hence, the time complexity of VAPOR remains linear. Furthermore, to efficiently implement matrix powers, we use block diagonal rotation matrices for $R$ as in Roformer (Su et al., 2024). VAPOR can be used similarly in the bidirectional setting.

## C EFFICIENCY

In this section, we focus on the runtime speed of the simple Latte layer as defined by Definition 1 and Definition 2 to eliminate other confounding factors like convolutions or sliding window attention. Figure 9 shows the computational and memory costs between standard causal attention and different versions of our algorithm for various sequence lengths. To isolate the complexity of the attention mechanism alone, we only calculate the time required to compute the transform $\tilde{x}$ of the input

---

[7]$e^{x_i} / \sum_i e^{x_i} = e^{x_i - x^*} / \sum_i e^{x_i - x^*}$, where $x^*$ is the maximum of the $x_i$ values.

tokens. Whilst Latte and other efficient attention mechanisms have better scaling properties with sequence length, GPU parallelisation of standard attention can make standard attention faster for short sequences, beyond which efficient approaches dominate. See Figure 10 for results on smaller sequence lengths. Since Latte time complexity is $O(TLD)$, the crossover point where it becomes faster than standard attention will increase with the number of latent states $L$, see Figure 12 in Appendix C.1. Hence, there is a tradeoff between the speed of the method and model capacity.

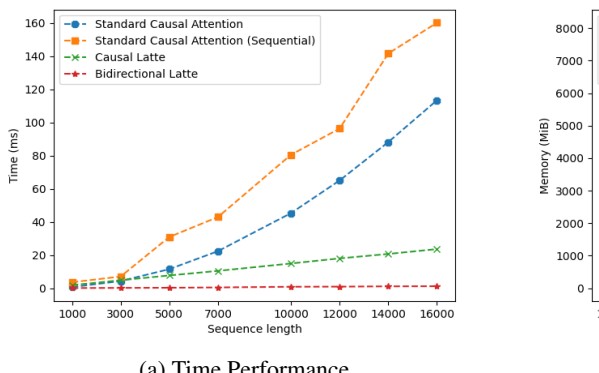

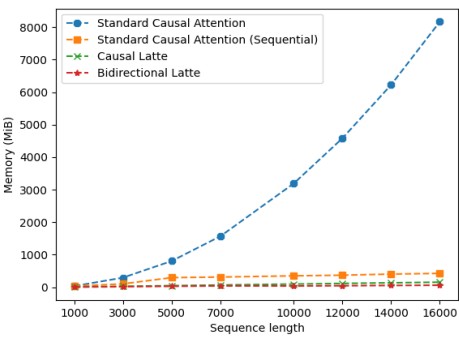

(a) Time Performance.

(b) Memory Performance.

Figure 9: Runtime comparison of Causal and Bidirectional Latte with both the Standard Causal Attention and the sequential implementation. Here we use batch size $B = 2$, transformer hidden dimension of $D = 128$, number of latent states $L = 128$ and number of heads $H = 4$. For all the sequential implementations, we unroll the loop 256 times (Appendix A.1.1) and group the operations into chunk sizes of 1000 for the sequential standard attention. We report results as an average of 100 repeats.

In Figure 10 we benchmark the time performance for Latte on smaller sequence lengths, complementing Figure 9a. We notice that Causal Latte is slower than Standard Causal Attention until sequence length 1600. Nonetheless, our implementation is always faster than the sequential version of Standard Causal Attention, with a smaller memory usage. The bidirectional case is faster and uses less memory for any sequence length.

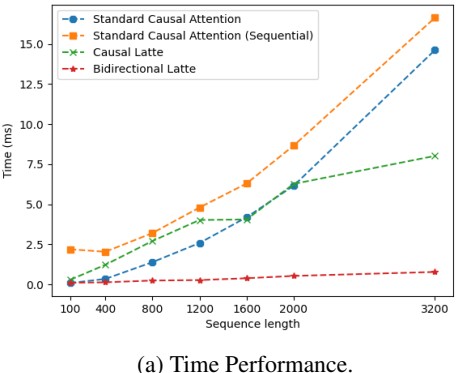

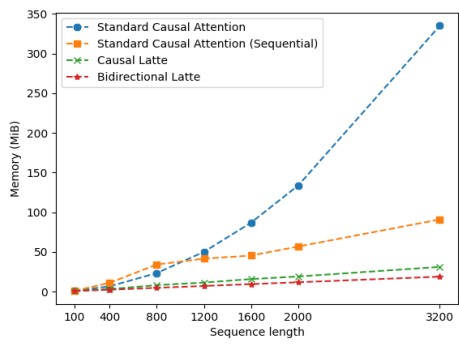

(a) Time Performance.

(b) Memory Performance.

Figure 10: Complementary version of Figure 9a for smaller sequence lengths. The same settings were used as in Figure 9a and we average the results over 100 runs.

We also do a comparison with FlashAttention (Dao et al., 2022) in Figure 11. We see that our linear mechanism is still faster on long sequences, even though we do not provide a CUDA hardware optimisation.

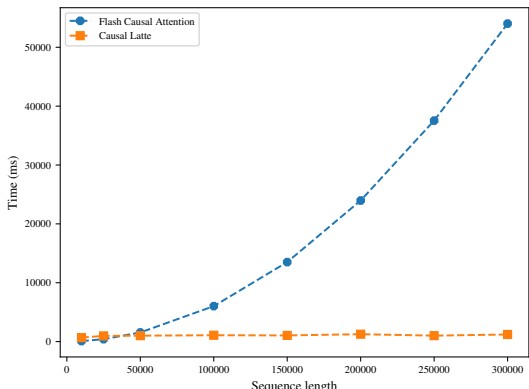

Figure 11: Flash Causal Attention compared to Latte Attention for one layer. Batch size is 5, head dimension is set to 8.

## C.1 Impact of Number of States on Runtime

We also analyse the runtime cost for different numbers of latent variables. We fix the sequence length $T$ to 5000, the hidden dimension $D$ to 512, the number of heads $H$ to 4 and batch size $B$ to 4, while we vary the number of latent states from 64 to 512. The loop unroll factor can be considered a hyperparameter concerning the computation time. In this case, we choose an unroll factor of 10. Figure 12 shows that both bidirectional and causal Latte is faster than the linear memory causal attention implemented with a scan; however, it is slightly slower than the standard causal attention when $L = D = 512$. This is explained by the fact that standard causal attention exploits parallelised matrix operations on GPU. Furthermore, as we previously showed in Section B.3, Latte is still faster when the sequence length increases.

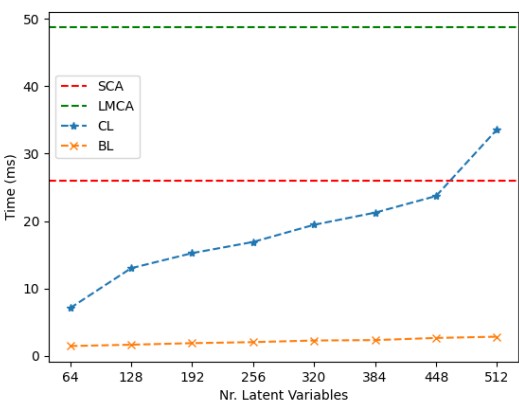

Figure 12: Run time versus the number of latent states $L$. SCA: Standard Causal Attention, LMCA: Standard Causal Attention (Sequential), CL: Causal Latte, BL: Bidirectional Latte. Experiment settings: $T = 5000, D = 512, H = 4, B = 4$

## C.2 Inference Performance

We take the models trained on OpenWebText data and study the inference time requirements for the language generation task. All the models receive an initial prompt of 20 tokens and generate sequences autoregressively using top 40 tokens sampling. The generated sequence length is 256

tokens. We run our experiments on both CPU and one A100 GPU and show in Table 10 that in the case of CPU-hosted inference and batch 16, our models are ten times faster than the transformer models. As we increase the batch size, the difference becomes even larger. On GPU, our models are still considerably faster than transformers for both batch sizes. Notably, we see that the runtime on GPU for Latte models is not impacted by the increase in batch size due to the size of the GPU.

Table 10: Inference time in seconds for models used in OpenWebText Table 1 for 256 sequence length.

| Model | Batch Size | CPU Time (s) | GPU Time (s) |
|---|---|---|---|
| Standard Att. (additive) | 16 | 807.318 | 399.290 |
| Rope (rotation) | 16 | 940.305 | 458.897 |
| Latte (additive) | 16 | 81.620 | 75.766 |
| VAPOR Latte | 16 | 89.163 | 101.935 |
| | | | |
| Standard Att. (additive) | 128 | 2002.587 | 554.828 |
| Rope (rotation) | 128 | 2228.802 | 622.909 |
| Latte (additive) | 128 | 111.778 | 75.223 |
| VAPOR Latte | 128 | 124.606 | 101.797 |

## D    RETRIEVAL CAPABILITIES

Linear models are known for being worse than transformers in retrieval capabilities(Arora et al., 2023). Hence we test the capabilities of our model on the synthetic MQAR data (Arora et al., 2023) and compare it with two other linear models and the standard transformer. Figure 13 shows that Latte-Mach++ performs competitively with the transformer and outperforms the other linear models in our training set. In all the experiments, the window size of attention is 128, being smaller than the context length.

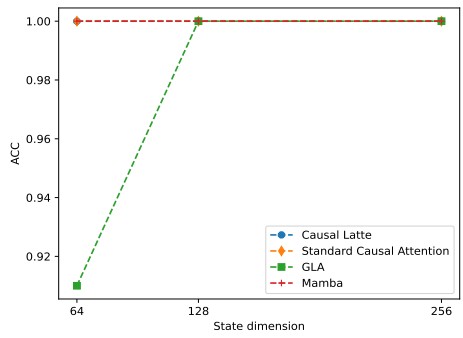
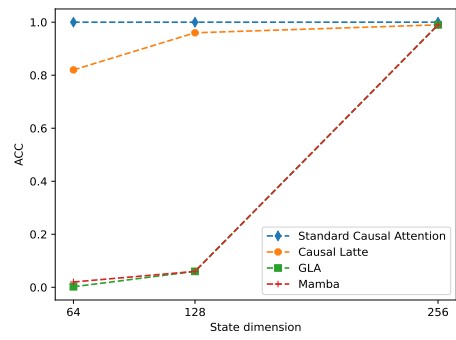

(a) Sequence length 256, number keys: 16          (b) Sequence length 512, number keys: 64

Figure 13: MQAR dataset on different sequence length and number of key-value pairs. We set the number of test examples to 10000 and train examples to 100000.

# E  CAUSAL LATTE IMPLEMENTATION

```python
@partial(jax.jit, static_argnums=(3, 5))
def causal_latte(Wq, Wk, Wv, H, X, unroll=100):
    """
    Scan implementation of latte.
    B: batch size H: nr heads, T: seq_len, D: hidden_dim. L: number latent states
    Args:
        Wq: jnp.array(DL), Wk:jnp.array(DL), Wv:jnp.array(DM) - parameter matrices
        H: int - nr heads
        X: jnp.array(BTD) - input
        unroll: int - unroll of the loop
    Returns:
        y: jnp.array(BTD) - transformed output sequence
    """

    def accumulate(carry, args):
        csum, norm_cumsum, prev_mx = carry
        Qs_t, curr_alph, V_t, c_mx = args
        revert_maxi = jnp.exp(-c_mx + prev_mx)
        add_maxi = jnp.exp(curr_alph - c_mx)

        norm_cumsum = jnp.einsum("BHL,BHL->BHL", norm_cumsum, revert_maxi)
        norm_cumsum += add_maxi
        carry = jnp.einsum("BHLD,BHL->BHLD", csum, revert_maxi)
        carry += jnp.einsum("BHL,BHD->BHLD", add_maxi, V_t)
        y = jnp.einsum("BHL,BHLD->BHD", Qs_t / norm_cumsum, carry)
        return ((carry, norm_cumsum, c_mx), y)

    B, T, D = X.shape
    L = Wk.shape[-1]

    V = jnp.einsum("DM,BTD->TBM", Wv, X).reshape(T, B, H, -1)
    Q = jnp.einsum("DL,BTD->TBL", Wq, X).reshape(T, B, H, -1)
    K = jnp.einsum("DL,BTD->TBL", Wk, X).reshape(T, B, H, -1)
    maxi = jax.lax.cummax(K, axis=0)

    init_alpha = jnp.zeros(shape=(B, H, L // H))
    init_carry = jnp.zeros((B, H, L // H, D // H))
    Qs = jax.nn.softmax(Q, axis=-1)
    _, y = jax.lax.scan(
        accumulate,
        unroll=unroll,
        init=(
            init_carry,
            init_alpha,
            K[0],
        ),
        xs=[Qs, K, V, maxi],
    )
    # TBHD -> BTHD
    y = y.transpose(1, 0, 2, 3)
    return y.reshape(B, T, D)
```

Listing 1: Scan version of Latte.

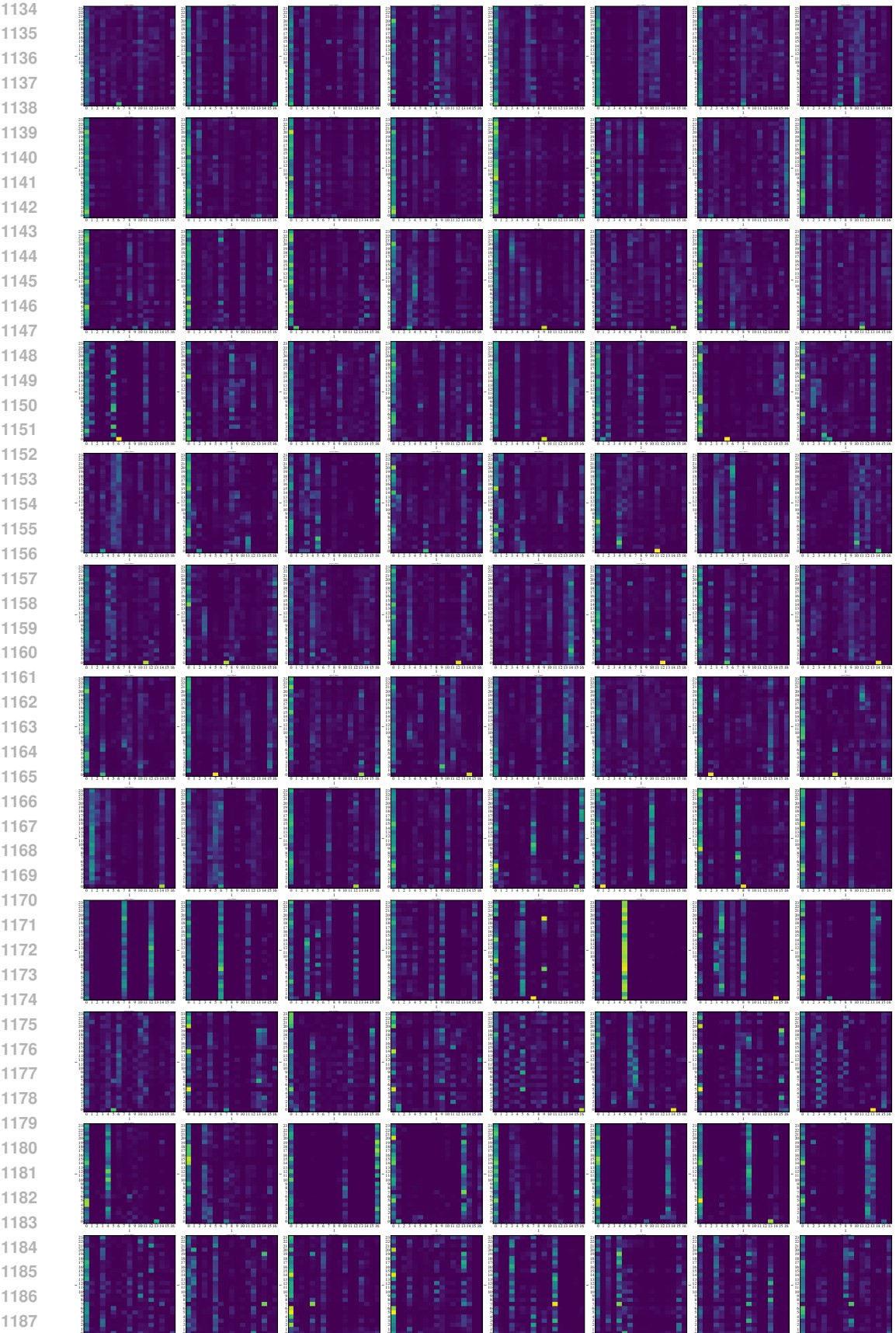

Figure 14: $p(l|t)$ for layers 1 to 12 and heads 1 to 8, from Section 4.3

