# OpenReview forum: "Latte: Latent Attention for Linear Time Transformers"
_ICLR.cc/2025/Conference — Submitted to ICLR 2025_

### Official Review · Reviewer_9nBN · 2024-10-21

**Soundness:** 1
**Presentation:** 4
**Contribution:** 1
**Rating:** 1
**Confidence:** 5

**Summary:**

The attention mechanism used in transformers has a time scaling of $O(N^2)$, where $N$ is the number of tokens. The reason for this quadratic scaling is due to the fact that the dot product between the query and keys is calculated for each pair of tokens when calculating softmax. The authors claim the presence of latent variables and show that by utilizing latent variables, they can approximate the softmax in $O(NL)$ time, where $L$ is the number of latent variables. They further improve their work by adding a sliding window attention, which calculates the original attention with respect to the neighboring tokens for each token. They evaluate their work by comparing the forward pass time against original attention, and the expressivity against original attention and a number of other transformers that have a timescaling of $O(N)$.

**Strengths:**

- The idea of replacing softmax$(QK^T)$ with softmax$(Q)$softmax$(K)^T$  is novel and has potential. However, the overall process is clustering (see weaknesses for more details).
- Including the literal code as an appendix is a good practice.

**Weaknesses:**

- "Latent Variable" is a very specific term used to describe the presence of a hidden variable that has a causal effect on the observed variables. The vectors you define as latent variable, $l$, are learnt so that the result of Equation 7 optimizes your objective function. In other words, the actual value of $l$ is determined by the specific values of the queries and keys, which themselves are derived by passing the tokens through a neural network. This makes the causal graph to be $x$ -> $(q,k)$ -> $l$. What your algorithm is doing is to perform a clustering with centers $l$ with respect to some objective.

- Regardless of the notion of transformers, if you are claiming the presence of a "latent variable", you need to perform proper causal inference techniques to prove your claim.

- Assuming the existence of latent variables, you should explain what these latent variables are supposed to represent. You should provide some intuition on why a hidden variable would be present, and what it might be. Otherwise, how could you argue about it's existence?

- The assumption of independence of $s$ from $t$ given $l$ in Definition 1 is not valid. The $s$ and $t$ are iterating over the same set of tokens. Each token $x$ directly affects the probabilities as written in Equation 6. Since they share a parent, they are correlated and not independent.

- Since the main motivation behind the design is improved time and memory complexity, the comparisons should've been made with FlashAttention [1]. Given that FlashAttention is calculating the softmax-based attention (regular attention) in an efficient manner, and is prevalent (arguably more prevalent the standard attention), it is not fair to compare your time and memory with the standard implementation of attention.

- In Equation 13, you're double counting the attention score for the nearby tokens of each token. Once through the sliding window and once by the Latte mechanism. This would: 1. over-emphasize on nearby tokens by assigning a higher score and 2. cause the sum of scores to be higher than $1$, voiding the mechanism of being a valid attention.

- In summary the overall idea has potential, but the math explaining why their method works is not sound. I encourage the authors to rewrite their paper with a clustering point of view, and include a time and memory comparison against FlashAttention and at least one of the other linear cost attentions.

[1] FlashAttention-2: Faster Attention with Better Parallelism and Work Partitioning

**Questions:**

- How dose "Latte Macchiato" compare with a transformer using just the sliding window attention? Specifically, a sliding window with a size of 128.

- Is the number of latent variables a hyper parameter, or is there a specific reason to choose them? i.e. does it scale with the input sequence length? Also, what is the number of latent variables in your experiments.

- In Figure 4, your forward pass scales sub-linearly with $N$. In fact it's almost constant. Why is the forward pass time not affected by the input sequence length?

- In Figure 7, you have mentioned that a benefit of Latte over other attentions with linear complexity is not collapsing. Could you elaborate what that means and why it would be troublesome. If collapsing is a well known phenomena you should add a citation.

---

> ### Author Response · Authors · 2024-11-21
>
> ## Weaknesses:
> 1.
>     -  The first point is incorrect. The definition of a latent variable is: “An unobserved quantity during train and test time”.  [1] Provides a good description of latent variable models and [2] provides a full derivation of examples like GMM which are latent variable models that are unrelated to causal inference.  Latent variable models and causality are two orthogonal concepts.
>    -   There is a general misunderstanding here. We now added the graphical representation of our model in Figure 8 (page 16). Note that by $p(s|t)$  we do not mean $p(x_s|x_t)$  which perhaps might be a misunderstanding from the reviewer.
>    -  All discrete latent variable models can be interpreted as clustering models. Naturally, our method performs a form of clustering in token embedding space, with the latent token states corresponding to cluster centres.  We assumed readers would be familiar with the well-recognised parallel between discrete latent variable modelling and clustering. However, we are happy to explain this point further if this would be useful to a reader.
> 2.
>     - This point is also incorrect. As mentioned previously, latent variables and causal inference are two independent notions. See [1] and [2].
>
> 3.
>     -  We mention that latent states represent global concepts on line 175. Latent states are associated with general concepts such as colours or shapes.
>     - Consider computing the attention between two tokens “red” and “green”. In standard attention, this similarity comparison goes directly via the token embeddings of these two concepts. However, one might argue that the reason “red” and “green” should be similar is because they are both colours. Therefore, in Latte, we know that “red” and “green” are similar since “red” is similar to the latent concept/token “colour”, and “green” is also similar to “colour”.
> 4.
>      -  This point is incorrect. Check the newly added Figure 8 from page 16. As mentioned previously our model consists of 2 discrete random variables s,t which denote positions and a discrete latent variable l. Under our graphical model, the assumption is correct.
> 5.
>       -  We compare it to standard attention and we consider it fair because Flash Attention is a Hardware optimisation of the standard attention. The time complexity of Flash Attention is still quadratic.
>      -  One can also provide hardware optimisation for Latte as in [3]. We are working on such a CUDA kernel, but it is not the main objective of our work and we feel would be a separate paper in itself. For example, Flash Attention is based on a careful and well-researched design decision, which is non-trivial. There is no in principle difficulty in creating a hardware-optimised version of Latte, but this will require some research and experimentation.
>     - We now added Figure 11 on page 19 which has a comparison with Flash Attention. The figure shows that FlashAttention is still quadratic.
> 6.
>     - This is wrong because we weigh the two attentions. Their sum is always one. The full attention is weighted by $p(l=0|t)$ and the latte attention is weighted by $p(l>0|t) $. Our attention scores always add up to 1 and are valid probabilities.
> 7.
>    -  Overall we strongly disagree with the reviewer. We suspect the reviewer’s belief in the lack of mathematical soundness points is based on a fundamental misunderstanding of the generally accepted definitions of latent variables and causality.
>     - We agree that Latte can be viewed as clustering and we can mention this in the paper. Indeed, this is a strength of latte and formed the basic motivation and intuition for the approach; we strongly disagree that our description based on using latent variable models is wrong and that we need to re-write the paper around clustering. However, we are happy to explain to readers the intimate relationship between clustering and discrete latent variable models.
>
> ## Questions:
> 1. We showed this in Figure 5.b
> 2. It is a fixed hyperparameter that we choose empirically. We have 128 latent states, information present in Tables 6 and 7 in the Appendix.
> 3. We modify the batch size as we increase the sequence length so that the number of tokens stays constant. We will improve our caption to explain this.
> 4. We do not say that there is a collapse of latent states in other linear attention models since other linear attention methods do not use the concept of latent variables. We only say that is generally possible in latent variable models to have latent collapse where only a few states are used. Then we show that this phenomenon does not occur in our model. We can explain this better if it would benefit the reader.
>
> [1] https://medium.com/@manasmahanta10/latent-variable-models-demystified-7f1342698985 \
> [2] Christopher M. Bishop. 2006. Pattern Recognition and Machine Learning\
> [3]  Yang, S., Wang, B., Shen, Y., Panda, R., and Kim, Y., “Gated Linear Attention Transformers with Hardware-Efficient Training”

---

### Official Review · Reviewer_JT5r · 2024-10-28

**Soundness:** 2
**Presentation:** 2
**Contribution:** 2
**Rating:** 5
**Confidence:** 4

**Summary:**

The authors propose Latte, a sub-quadratic attention alternative based on a low-rank re-parameterization – a latent state is defined, and input tokens attend to this latent to break the quadratic dependence on sequence length. Additionally, a hybrid model that combines local standard attention and Latte is proposed to improve local processing.

For language modeling, small-scale pre-training experiments on 8B tokens of OpenWebText are conducted, and the proposed attention alternatives are compared to other attention variants in terms of test perplexity.  In addition, the model is compared to other efficient attention ops on Long Range Arena (LRA), and up-training experiments to extend Gemma 2.6B by replacing attention with Latte-Macchiato  are conducted.

**Strengths:**

- Though many linear attention variants have been proposed over the past two years, the approach in this paper appears novel
- It compares favorably on small-scale PPL and reasonably on LRA evaluations compared to other recent proposals
- The Latte operation is well-motivated and clearly derived
- The experimental transparency on hyperparams, training code and experimental details is commendable

**Weaknesses:**

The primary weakness of this paper is the experimental evaluation -- it is unclear from the experiments in this paper the extent to which the results would extend to natural-language and long-context evaluations.  A number of prior works (T2R [1], Hedgehog [2] and SUPRA [3] which are missing in the discussion in Section 4.5) take pre-trained vanilla attention Transformers and fine-tune / adapt them to linear and efficient alternatives.  The findings in the SUPRA study [3] show that there are gaps between efficient attention models and standard attention for long-context tasks.  Thus proper comparisons on natural language evaluations (Hellaswag, ARC, etc.) and long-context (Scrolls [4]) tasks would illuminate the strength of Latte/Latte-Macchiato vs standard attention.

Others:
- In Figure 6, is the context length for vanilla attention Gemma extended using the YaRN [5] trick? Since there is a standard fine-tuning free approach that is now commonly used when context length exceeds pre-training context, it should be used for fair comparison with vanilla attention

[1] Kasai, Jungo, Hao Peng, Yizhe Zhang, Dani Yogatama, Gabriel Ilharco, Nikolaos Pappas, Yi Mao, Weizhu Chen, and Noah A. Smith. "Finetuning pretrained transformers into rnns." EMNLP 2021

[2] Zhang, Michael, Kush Bhatia, Hermann Kumbong, and Christopher Ré. "The hedgehog & the porcupine: Expressive linear attentions with softmax mimicry." ICLR 2024

[3] Mercat, Jean, Igor Vasiljevic, Sedrick Keh, Kushal Arora, Achal Dave, Adrien Gaidon, and Thomas Kollar. "Linearizing Large Language Models." COLM 2024

[4] Shaham, Uri, et al. "Scrolls: Standardized comparison over long language sequences." EMNLP 2022

[5] Peng, Bowen, Jeffrey Quesnelle, Honglu Fan, and Enrico Shippole. "Yarn: Efficient context window extension of large language models." arXiv preprint arXiv:2309.00071 (2023).

**Questions:**

Larger-scale experiments to fully validate Latte/Latte-Machiatto compared to vanilla attention may be expensive, but the finetuning experiments already conducted in Section 4.5 may be suggestive of natural language performance.

At the 2.6B scale there should be signal in terms of standard natural language and long context evaluations -- running the trained Gemma-Macchiato on the standard harness of natural language (Hellaswag, MMLU, etc.) and natural language long-context evaluations (Qasper, NarrativeQA, etc.) would go a long way toward verifying that performance is maintained with the base model at short context, and improves at long context tasks.  It may also be interesting to conduct these experiments with the other proposed Latte variants.

Especially interesting would be results of these experiments on MMLU, where linear attention variants have struggled (as in SURPA [3] above, where the gap between the base and linearized models are small on Hellaswag but very large on MMLU).

---

> ### Author Response · Authors · 2024-11-21
>
> ## Weaknesses:
> 1.
>   -  We now introduced experiments in Table 5 showing the results of the pre-trained model on natural language harness (MMLU, Hellaswagm, ARC).
>   - Compared to T2R and Hedgehog our method has a few advantages:
>        - Unlike T2R and SUPRA we have local attention which helps improve results as shown by experiments in  Table 1 and Table 2.
>        - We do a weighted combination of sliding window attention and our Latte attention. Hence we can freeze all the parameters of the network relating to sliding window attention (and the MLPs of the transformed) and adapt only the parameters of  Latte attention. This means that we can readily take an off-the-shelf model and readily merge this with our Latte model (resulting in the Latte-Macchiato approach). This reduces the memory required by the optimizer and the training speed.
>        - Hedgehog requires quadratic training cost since it needs to compute the full attention in its training objective, hence it cannot be trained on long sequences. One can apply the methodology in HedgeHog to our model as well and this would make for interesting future work. We see these as independent contributions.
>    - We thank the reviewer for suggesting these related works, some of which we were unaware of and appeared since we began our research. We will cite them.
>    -  We see our work as part of the major effort to make linear sequence models as performant as standard quadratic attention transformers. While the problem is not completely solved we tackle some issues by introducing a hybrid model which uses a valid probability distribution for attention.
>
> 2.
>    - No, we do not use Yarn. Methods like Yarn or Xpos can extend the context of a pre-trained transformer, but they maintain its quadratic complexity. Conversely, our model extends the context at only linear scaling complexity.

---

> > ### Comment · Reviewer_JT5r · 2024-11-22
> >
> > The new Table 5 is helpful, is this the model trained for 8B tokens of OpenWebText?
> > In terms of Yarn, I meant that this kind of technique is now standard for extending context beyond training context and since many variants are training-free, for a fair comparison it's good to include since it allows most models to extrapolate well beyond their training seqlen.

---

> > > ### Author Response · Authors · 2024-11-23
> > >
> > > In table 5 we use the model trained on top of Gemma on the SlimPajama dataset.
> > >
> > > In Figure 6, for a fair comparison, we replaced ROPE with YARN in the standard causal attention experiment. However, without performing additional training when sequences get longer, the YARN model still does not extrapolate as well as ours. The only difference is that the perplexity gets lower with YARN than with ROPE (at least in our book corpus experiment from Figure 6 where we train from scratch).
> > >
> > > We do not think that it would be fair to do additional training for YARN as in this experiment we do not do additional training for Latte Macchiato when the sequence length increases.

---

### Official Review · Reviewer_sTmJ · 2024-11-02

**Soundness:** 2
**Presentation:** 2
**Contribution:** 2
**Rating:** 3
**Confidence:** 5

**Summary:**

This paper introduces a linear complexity attention mechanism for sequence modeling. The central concept involves processing the Q and K matrices with the softmax operation independently. Additionally, the paper explores mixed attention through sliding window attention, demonstrating enhanced performance in language modeling. Experimental results on both language modeling and LRA tasks indicate competitive performance. The paper also presents distillation results using pre-trained models.

**Strengths:**

The paper is well-written.

**Weaknesses:**

1. The concept is quite similar to Efficient Attention: Attention with Linear Complexity. Although the author clarifies the differences, such as from vision tasks to language modeling and latent variable interpretation, I believe the novelty is still limited. First, vision tasks are a 2D sequence modeling problem, which is more complex than a 1D language modeling problem. Second, the latent variable interpretation treats the Q and K matrices as attention matrices, which seems a bit strange to me.

2. There is a significant lack of linear models for comparison in this context. For instance, models such as HGRN (NeurIPS), HGRN2 (COLM), Lightning Attention (ICML), and GLA (ICML) are missing. Additionally, it is well-known that linear models may perform well on a small scale but often fail to scale effectively. The experiments conducted with 150 million parameters are insufficient to validate the actual scaling capabilities of the proposed method. Furthermore, the distillation results do not provide evidence of these scaling capabilities.

3. Is the standard causal attention implemented with flash attention or not for the speed comparison? If not, the comparison results are not helpful. Also I would suggest include sota linear attention variants for comparison as well.

4. It is well known that the limitation of linear models is their retrieval capability. The paper lacks experiments on "Needle in a Haystack" to demonstrate its performance on long sequence modeling.

**Questions:**

As above.

---

> ### Author Response · Authors · 2024-11-21
>
> ## Weaknesses:
> 1.
>   - Our work is novel because, as far as we know, it is the only causal linear attention model that has the intuitive interpretation of defining latent states (corresponding to token clusters) whilst retaining a correctly normalised attention distribution.
>   - Furthermore, our model has natural formulations for both the bidirectional and causal cases and provides an extension to hybrid models that integrate standard sliding window attention with linear attention, retaining correctly normalised attention throughout. As far as we know, this is the only model with such features. Efficient attention cannot be used to model the causal case nor can it be combined with sliding window attention.
>   - The Q and K matrices are related to attention because our model effectively computes attention between tokens of the sequence and latent (token) states. An intuitive example is to think about each latent state as a general concept like shape or colour. Consider computing the attention between two tokens “red” and “green”. In standard attention, this similarity comparison goes directly via the token embeddings of these two concepts. However, one might argue that the reason “red” and “green” should be similar is because they are both colours. Therefore, in Latte, we know that “red” and “green” are similar since “red” is similar to the latent concept/token “colour”, and “green” is also similar to “colour”. This is the essential intuition behind Latte — that similarity between tokens can be more succinctly expressed by similarly between tokens and learned latent tokens.
>
> 2.
>     -  There are in-depth comparisons with other sequence models in Table 2 and, at the reviewer’s suggestion, we further introduced GLA and Lightning Attention, showing that Latte-Macchiato outperforms those approaches.
>    -  As a humble university research department, we do not have the compute necessary to run large-scale experiments and the experiments we have presented are already at the limit of our computational resources. We provided comparisons of Latte with other linear models showing the potential of our idea. We hope that others with larger computational resources can be inspired by these results and explore the scaling of Latte to larger models. Latte is, we believe, a remarkably simple and easily implementable drop-in replacement for standard attention.  The simplicity of Latte means that it can also be easily integrated with standard sliding window attention, as we have shown with Latte-Macchiato.
>     -  We understand the reviewer's concern that our distillation results don’t necessarily provide evidence of scaling in terms of pre-training. However, we strongly believe that our results show evidence of Latte-Macchiato working at scale. Our distilled model is a 2.7B parameter Gemma model extended with Latte-Macchiato and we show in Table 4, that it outperforms the original model on autoregressive language modelling. We believe this provides clear evidence of Latte’s scaling capabilities. Naturally, we would like to apply the same approach to much larger base models, but this is currently beyond our computational resources.
>
> 3.
>      -  No. Originally, we only compared it with standard causal attention (without Flash Attention) because Flash Attention is a hardware optimisation of standard attention. The time complexity of Flash Attention is still quadratic and therefore will be slower than linear attention beyond a certain sequence length. The crossing point is naturally dependent on hardware optimisation. Our work is not yet hardware-optimised; we are working on a CUDA kernel, but it is not the main goal of this paper and (like Flash Attention) would be a separate research contribution.  However as requested, we have now included a comparison with Flash Attention in Figure 11, Appendix C and show that for long sequences Latte is faster.
>    -   All linear methods have linear slopes in terms of time versus sequence length, albeit with potentially different slopes.  This means that all linear methods (including Latte) will eventually outperform any quadratic attention (even those that are hardware-optimised). For this reason, we do not feel it useful to plot the scaling of all other linear attention mechanisms we considered.
>
> 4.
>     - This is a very good suggestion. We now incorporated MQAR [Arora] synthetic dataset in our experiments and show in Appendix D that, for our settings, the model performs better than other linear models. We use a window of size 128 for full attention, which is smaller than the entire context. Whilst our method is, we believe, state-of-the-art for linear scaling methods, finding a linear scaling attention approach that preserves excellent performance on needle-in-a-haystack problems remains an elusive goal for the research community.

---

### Official Review · Reviewer_rTYN · 2024-11-03

**Soundness:** 2
**Presentation:** 3
**Contribution:** 2
**Rating:** 5
**Confidence:** 4

**Summary:**

This paper aims to alleviate the well-known problem of Transformers -- quadratic complexity. The main idea of this paper is instead of storing all KV cache, it adopts a fixed number $L$ of latent tokens with the goal of embedding global information into the fixed number of states. By having a fixed number of the latent tokens, it has a fixed computational complexity that's independent of input sequence lengths. The authors also present an efficient causal update mechanism which is significantly important during inference. Finally, by adding additional techniques such as sliding window attention and RG-LRU, the proposed module shows competitive performance comparing to the vanilla self attention while maintaining efficiency.

**Strengths:**

The paper is very easy to follow and the derivation of its bidirectional and causal forms is succinct. It also shows various connections to many different previous works such as Vanilla attention, SSMs, Linear attention, and etc. By cleverly reformulating the algorithm, the module can be adapted to the causal setting, which is significantly important lately due to auto regressive training. The paper demonstrates its competitiveness in diverse settings.

**Weaknesses:**

- Dependence on SWA and LRU. Compared to recent SSMs like Mamba, it doesn't have a clear advantage given that it is dependent of other methods and slower inference.
- While the authors mention that the implementations of Luna and Latte differ substantially, it is unclear how they are fundamentally different. Without the additional techniques that Latte integrates such as SWA and LRU, it is uncertain whether Latte clearly has substantial performance-wise benefits over Luna. If so, why is there a mathematical reason?
- Although the causal variant is efficient during inference, parallel training requires Jax framework, which again hinders independence of this method from other settings.
- For a fair comparison, I believe previous models that can be used as a counterpart of Latte should be reevalated by replacing Latte with those models and have other components like SWA++ and RG-LRU the same.

typo:
- line 69 : $\sum^T$ to $\sum^t$

**Questions:**

- Are the runtime results for transformers measured using FlashAttention [A]?
- The sequence extrapolation is very interesting, but I wonder if it is mainly due to sliding window attention. Does Latte itself also extrapolates well?

[A] Dao et al. FlashAttention: Fast and Memory-Efficient Exact Attention with IO-Awareness

---

> ### Author Response · Authors · 2024-11-21
>
> ## Weaknesses
> 1.
>    -  An advantage of our method over Mamba and other Linear Attention models is that it preserves the probabilistic properties of attention. This has intuitive appeal and practical benefits such as correct normalisation over sequences and the fact that we preserve some interactions between tokens through our latent variables.
>    - The reviewer is correct that Latte-Macchiato (the extension of Latte) is dependent on SWA. However, we believe this to be a key advantage of our approach as we combine both linear and SWA attention with correct normalization such that their summation remains a valid probability distribution over tokens. We believe this is important to avoid overestimating tokens in the window over tokens outside of the window.  We thank the reviewer for raising this point and we will make sure to amend this in our updated paper.
>   -  Our method does not have slower theoretical inference —  it is still constant next token prediction. Our model also supports hardware optimisation — however, hardware optimisation is not the main aspect of our paper and will be done in future extensions.
>
> 2.
>   -  We thank the reviewer for raising this point as we believe that our mathematically sound integration of SWA with Latte is a key advantage over models such as Luna. Indeed, models such as Luna do not support the combination of local standard attention and linear attention, whereas in Latte we combine linear attention with SWA whilst ensuring a valid probability distribution to attend over token indices.
>   - Luna has never been applied to autoregressive language modelling. In Table 2 we perform several of our own comparisons with other linear models and show that Latte achieves state-of-the-art performance on autoregressive language modelling. We also added a new experiment in Appendix D showing that our model is better at retrieval tasks than other linear models.
>
> 3.
>   -  We would like to stress that our model does not require Jax and is agnostic to whichever machine learning library one might prefer. We chose Jax for simplicity given the pre-existence of the “scan” function, although we stress again that this could be implemented in PyTorch if the user so wished.
>
> 4.
>   - While such a comparison would be interesting, checking all possible model configurations is beyond the scope of our paper. In our paper, we performed a thorough investigation of the benefits of different components and believe that the novelty of our work comes from how we preserve a valid probability distribution over token indices while combining standard causal attention and linear Latte attention. We do not know of any other linear sequence models that can be combined with standard attention while preserving a probability distribution.
>
> 5.
>    - Thanks, we fixed the typo.
> ## Questions:
> 1.
>   -  No. Originally, we only compared it with standard causal attention (without Flash Attention) because Flash Attention is a hardware optimisation of standard attention. The time complexity of Flash Attention is still quadratic and therefore will be slower than linear attention beyond a certain sequence length. The crossing point is naturally dependent on hardware optimisation. Our work is not yet hardware-optimised; we are working on a CUDA kernel, but it is not the main goal of this paper and (like Flash Attention) would be a separate research contribution.  However as requested, we have now included a comparison with Flash Attention in Figure 11, Appendix C and show that for long sequences Latte is faster.
>
> 2.
>     - Yes, we find that Latte also extrapolates. Since Latte Macchiato combines standard sliding window attention and our latte attention, the excellent extrapolation properties of Latte Macchiato derive from both SWA and the Latte mechanism itself.

---

### Meta-Review · Area_Chair_ka51 · 2024-12-20

**Metareview:**

All reviewers voted for rejecting the paper post rebuttal. The AC checks all the materials, and while appreciating the efforts on clarifications/new results and agree with some points in the rebuttal, the AC finds the reviewer consensus more appealing, and decides that the paper can be further improved and would benefit from another cycle. The authors are encouraged to make changes according to the reviews and submit this work to the next venue.

**Additional Comments On Reviewer Discussion:**

Notes from the reviewer discussion:
- Runtime measured by standard self-attention implementation vs. Flash Attention. This is a shared concern from reviewers. The AC agrees with the authors that theoretically, Flash Attention still preserves the quadratic complexity from standard self-attention, but also think it's at least better to include runtime results from Flash Attention *as a reference* in the main paper. It does not diminish the contribution and readers will appreciate the transparency.
- The results on MQAR synthetic dataset for retrieval is a valuable addition. To the AC, this can also be one of the main-paper results, though some modifications are needed for the presentation (e.g. the legend and ordering of methods between the two figures shall be consistent).
- Quite a few writing points are promised by the authors (e.g. probability distribution over tokens as a key property of the proposed method), and have not yet been incorporated in the most recent draft. The authors are highly encouraged to update and improve the draft accordingly.

---

### Decision · Program_Chairs · 2025-01-22

Reject